# Psychosocial Profiles of Men Who Have Sex with Men (MSM) Influencing PrEP Acceptability: A Latent Profile Analysis

**DOI:** 10.3390/bs15060818

**Published:** 2025-06-14

**Authors:** Anthony J. Gifford, Rusi Jaspal, Bethany A. Jones, Daragh T. McDermott

**Affiliations:** 1NTU Psychology, School of Social Sciences, Nottingham Trent University, Nottingham NG1 4FQ, UK; beth.jones@ntu.ac.uk (B.A.J.); daragh.mcdermott@ntu.ac.uk (D.T.M.); 2Vice-Chancellor’s Office, University of Brighton, Brighton BN2 4GJ, UK; r.jaspal@brighton.ac.uk

**Keywords:** LPA, PrEP, MSM, UK, acceptability, uptake

## Abstract

Despite the availability of pre-exposure prophylaxis (PrEP) in the United Kingdom (UK), uptake among men who have sex with men (MSM) remains inconsistent, signalling a persistent ‘PrEP Gap’. Empirical studies show the important role of psychosocial factors (e.g., stigma, identity, trust in science, and sexual behaviours) in PrEP acceptability and uptake. This study aimed to identify subgroups of MSM in the UK based on psychosocial predictors of PrEP acceptability. A cross-sectional survey of MSM (*N* = 500) was conducted between June and September 2023. Participants completed validated measures assessing identity resilience, internalised homonegativity, LGBTQ+ connectedness, trust in science, NHS perceptions, HIV stigma, PrEP self-efficacy, condom self-efficacy, sociosexual orientation, perceived HIV risk, and PrEP acceptability. Latent Profile Analysis (LPA) was used to identify distinct subgroups based on these psychosocial dimensions. Four psychosocial profiles were defined: (1) PrEP Ambivalent (15%); (2) PrEP Accepting (36.2%); (3) PrEP Hesitant (37%); and (4) PrEP Rejecting (11.8%). These profiles provide evidence for varied combinations of personal and structural factors influencing PrEP acceptability. PrEP acceptability among MSM in the UK is shaped by distinct psychosocial configurations, influenced by identity, stigma, trust, and perceived risk. These findings highlight the need for differentiated and targeted interventions for enhancing PrEP acceptability based on psychosocial profile. Audience segmentation strategies offer a promising pathway to bridge the awareness-to-engagement gap and address the nuanced barriers facing diverse subgroups within the MSM community.

## 1. Introduction

Ending the human immunodeficiency virus (HIV) epidemic remains a vital public health goal for the United Kingdom (UK) ([75]). This includes the ambitious target of eliminating HIV transmission by 2030 ([1]). However, the latest HIV surveillance data indicate a rise in new infections in the UK, with a 15% rise in 2023 ([58]). Furthermore, despite HIV rates slowly decreasing for men who have sex with men (MSM), this decline is starting to plateau. As such, efforts must be maintained to help those most at risk of HIV adopt protective behaviours ([9]). The implementation and continued accessibility of pre-exposure prophylaxes (PrEP) through the National Health Service (NHS) is an effective public health intervention to achieve this goal ([60]).

PrEP is currently available in the UK free at the point of access, administered as an oral medication of emtricitabine/tenofovir disoproxil and typically taken daily ([5]; [56]). Clinical trials, including the PROUD study conducted in the UK, have demonstrated that oral PrEP is highly effective at reducing HIV acquisition among populations at increased risk of HIV acquisition ([20]; [60]). PrEP uptake and adherence are imperative for this intervention to be effective at an epidemiological level ([57]). However, despite free access to PrEP, research shows a significant ‘PrEP Gap’ among populations considered clinically to be at elevated risk of HIV infection (e.g., gay and bisexual MSM; [70]). Physical barriers (e.g., poor access to sexual health clinics) may impact access to PrEP ([16]; [61]). Additionally, social and psychological factors, such as stigma and mistrust in science, can undermine PrEP uptake ([39]; [49]; [51]). These psychosocial barriers can lead to hesitancy, avoidance, or outright rejection of PrEP, even among those who are clinically eligible and aware of its availability and efficacy.

PrEP also offers significant psychological benefits, such as reduced HIV anxiety ([51]). It can further contribute to feelings of prosocial behaviour, as it serves to protect not only those who use PrEP, but also their sexual partners ([48]). Thus, PrEP uptake and acceptability are not driven solely by perceived risk of HIV, if at all (e.g., [18]), but also by a range of motivational and attitudinal factors that shape how individuals engage with sexual health interventions ([92]). These include personal beliefs, identity-related factors, and psychological idiosyncrasies. However, limited research has explored these individual or ‘user-level’ factors in depth, particularly examining how they influence decisions to accept or reject PrEP, as described by [39] ([39]). Further research is needed to examine how the uptake or rejection of PrEP is shaped by underlying psychosocial factors and, by extension, the psychological characteristics of those at risk of contracting HIV.

To support broader implementation of PrEP across the UK, it is increasingly recognised that new approaches are needed, especially those that move beyond simply raising awareness, to actually fostering meaningful engagement ([29]). This approach requires the gap from ‘awareness’ to ‘engagement’ to be addressed ([30]). Studies applying behaviour change frameworks have highlighted that psychological barriers to PrEP uptake vary across populations with elevated risk of HIV acquisition, suggesting that different subgroups may require tailored intervention strategies ([27]). However, much of this work has relied on small-scale qualitative studies. This shows a growing need for larger-scale quantitative studies to help triangulate this research across a broader population span.

In response, there is growing interest in the use of audience segmentation, a technique from market research that recognises population level heterogeneity and divides a population into homogenous subgroups based on shared characteristics to inform targeted health communication. While segmentation strategies have traditionally focused on demographic traits, recent literature has begun to advocate for segmentation based on psychometric and psychosocial factors, particularly in the context of PrEP ([19]; [71]; [74]; [81]). These approaches allow health messaging to be tailored more precisely, enhancing relevance and impact. Such techniques have been applied widely in HIV/AIDS awareness and prevention efforts ([76]; [77]; [87]) and are now gaining traction in campaigns to increase PrEP uptake among specific target demographics.

The selection of psychosocial variables in this study is grounded in both emerging empirical work (e.g., [35]) and established findings across the HIV prevention literature (e.g., [16]; [39]; [48]). Constructs, such as internalised homonegativity and LGBTQ+ connectedness, have been found to shape both health-related decision-making and engagement with sexual health services among MSM ([24]; [50]). Additionally, trust in science and attitudes toward the NHS reflect broader institutional and cultural influences that can impact PrEP adoption, particularly in the context of medical mistrust or perceived exclusion from healthcare systems ([11]). Variables such as perceived HIV risk, stigma, condom self-efficacy, and sociosexual orientation have also consistently emerged as key predictors of sexual health behaviours, including PrEP uptake and acceptability ([8]; [10]; [43]). Collectively, these variables were selected to capture a multidimensional view of the psychological, social, and structural factors likely to influence engagement with PrEP.

Given the established relationship between psychosocial barriers and PrEP uptake and acceptability among MSM ([12]; [39]; [48]), this current study uses Latent Profile Analysis (LPA) to investigate the association between psychosocial dimensions and PrEP acceptability. While such factors influencing PrEP uptake have been widely researched, there remains a need for larger-scale, quantitative, person-centred analyses in the UK, particularly in the context of free access to PrEP. This study seeks to identify possible PrEP-user subgroups within a sample of MSM individuals to examine distinct contributing factors to PrEP acceptability—a measure that can help predict the likelihood of PrEP use ([53]). This allows for further understanding of how PrEP usage and uptake may differ across MSM populations in the UK ([35]). This approach not only builds on existing research but also provides critical insights to inform tailored public health interventions and bridge the persistent PrEP Gap in the UK. Specifically, the following research questions are explored:Are there heterogeneous subgroups of MSM in relation to psychosocial factors and PrEP acceptability?Can these subgroups (if any) be qualitatively defined to provide scope for targeted health communication interventions?

## 2. Method

### 2.1. Participants and Recruitment

Gay, bisexual, and other men who have sex with men (MSM) were recruited to take part in an online cross-sectional survey study between June—September 2023. Participants were eligible if they were aged 18 or over, were assigned male at birth, lived in the UK, self-identified as gay, bisexual, or men who have sex with men, and were not undergoing any gender affirming care. Participants were recruited using opportunistic sampling through social media (e.g., Twitter/X, Instagram, Facebook).

The survey took on average 18 min to complete, with attention checks throughout. A total of 903 participants consented to complete the survey. Of these, *N* = 391 failed to complete the survey or failed the ‘bot-checker’ built into the online survey platform ([62]). A further *N* = 12 participants were removed for not meeting the eligibility criteria or failing attention checks. A participant group of *N* = 500 was used for the final analysis.

This study received a favourable ethical opinion from the Schools of Business, Law and Social Sciences Research Ethics Committee of Nottingham Trent University. 

### 2.2. Measures

Psychometric measures were selected following the conventions of similar LPA studies in this area (e.g., [79]), after synthesising the literature on current psychosocial barriers to PrEP uptake. Additionally, context-specific research (e.g., [29]; [35]) further highlighted key constructs warranting additional exploration (i.e., internalised homonegativity, LGBTQ+ connectedness, trust in science) that align with our study’s aim of generating hypotheses about how these psychosocial factors influence PrEP acceptability. Each measure was therefore chosen for its theoretical relevance and empirical support in relation to our overarching research questions.

The following sociodemographic data were collected: age, sexuality, gender identity, relationship status, marital status, ethnicity, education, location in the UK, residency (i.e., city centre/urban, countryside/rural, suburban). Demographic data of specific importance to the study pertaining to the participant’s profile were also collected: outness, HIV status, current or past PrEP usage, and PrEP dosing mechanism (e.g., daily PrEP).

### 2.3. Identity Resilience

The Identity Resilience Index ([6]) is a 16-item measure providing an overall score of identity resilience (comprised of self-esteem, self-efficacy, positive distinctiveness, and continuity). Example items include ‘On the whole I am satisfied with myself’ (self-esteem) and ‘I can always manage to solve difficult problems if I try hard enough’ (self-efficacy), measured on a five-point Likert scale (1—strongly disagree, 5—strongly agree). Higher mean scores indicate higher identity resilience. This scale had excellent internal reliability in related research (α = 0.83; [7]).

### 2.4. Internalised Homonegativity

The Revised Internalised Homophobia Scale ([44]) is a five-item measure. Wording was adapted to suit male participants. An example item is ‘I have tried to stop being attracted to men in general’ and items were measured using five-point Likert scale (1—strongly disagree, 5—strongly agree). Higher scores indicate a higher level of internalised homonegativity and in a related study this had excellent internal reliability (α = 0.88; [50]).

### 2.5. LGBTQ+ Connectedness

The Connectedness to the LGBT Community Scale ([32]) is an eight-item scale. An example item is ‘You feel a bond with the LGBTQ+ community’ (the acronym was expanded to include ‘Queer’). It was measured using a four-point Likert scale (1—strongly disagree, 4—strongly agree) with a higher score indicating a higher sense of connectedness. When validated this scale had excellent internal reliability (α = 0.81; [32]).

### 2.6. Trust in Science

A shortened version of Trust in Science and Scientists Inventory was used ([65]; [7]). This is a 12-item scale measured on a five-point Likert scale (1—strongly disagree, 5—strongly agree). An example item is ‘Scientists ignore evidence that contradicts their work’. Lower scores indicated a higher trust in science. Other research using the abridged scale reported good internal reliability (α = 0.83; [7]).

### 2.7. NHS Attitudes

Perceptions of how the National Health Service (NHS) serves a participant’s community was measured by taking three items from the NHS Experience Scale ([31]). The starting clause was ‘thinking about the community you see yourself as part of, do you think…’ and an example item is ‘your community is looked after by the NHS?’. Possible answers were ‘No’ (1), ‘Maybe’ (2) and ‘Yes’ (3) and a higher score indicates a more positive perceptions of the NHS.

### 2.8. Perceived Risk of HIV

The Perceived Risk of HIV Scale ([67]) is eight-item scale measuring HIV risk specific dimensions such as likelihood estimates of infection, intuitive judgements, and salience of risk. Items are measured on a variety of Likert scales for example ‘what is your gut feeling about how likely you are to get infected with HIV?’ (1—extremely unlikely, 5—extremely likely). Higher scores indicate a higher perceived risk of contracting HIV. When validated this scale had excellent internal reliability (α = 0.88; [67]).

### 2.9. HIV Stigma

An adapted ‘Stigmatising Attitudes towards People Living with HIV/AIDS Scale’ was used ([3]). The English version of the scale was used, and the word ‘AIDS’ was amended to HIV to reflect cultural and medical changes in HIV discourse ([91]). One item ‘it would not bother me if there was sheltered accommodation for people with HIV on my street’ was adapted to be more applicable to a UK sample. It is a 27-item scale, measured on a five-point Likert scale (1—strongly disagree, 5—strongly agree). The items are reverse-coded so that a higher score indicates a lower stigmatising attitude towards HIV. In a related study, this had excellent internal reliability (α = 0.92; [52]).

### 2.10. Willingness to Sleep with Someone HIV+ Undetectable

A single-item measure to understand a likelihood of sleeping with someone who is HIV+ with an undetectable viral load was used ([21]). We asked participants to consider if ‘I am willing to have anal sex without condoms with a HIV-positive partner who has an undetectable viral load’ and is measured on a six-point Likert scale (1—very strongly disagree, 6—very strongly agree). A higher score indicates a higher likelihood of having sex with someone living with HIV with an undetectable viral load.

### 2.11. Risky Sexual Behaviour

The Sexual Risk Behaviours Scale ([28]) is a five-item scale measured on a five-point Likert scale (1—never, 5—very often). An example item is ‘how often have you had sex without a condom with someone you have just met?’ Higher scores indicate higher frequencies of risky sexual behaviour. When validated this scale had excellent internal reliability (α = 0.84; [28]).

### 2.12. Attitudes Towards Uncommitted Sex (Socio-Sexuality)

The revised Sociosexual Orientation Inventory ([72]). This scale captures three theoretically meaningful components of sexual desire using nine items. Example items include ‘with how many partners have you had sex within the last 12 months?’ (past behavioural experiences); ‘sex without love is ok’ (attitude towards uncommitted sex); and ‘in everyday life, how often do you have spontaneous sexual fantasies about having sex with someone you have just met?’ (socio-sexual desire). Items were measured on various nine-point Likert scales, and an amalgamated score was calculated. Higher scores represent a more accepting attitude and desire for uncommitted sex. In a related study this had excellent internal reliability (α = 0.84; [93]).

### 2.13. Condom Self-Efficacy

The Brief Condom Use Self-Efficacy Scale ([37]) with pronouns adjusted to be gender neutral (i.e., he/she > they) was used. This is a seven-item scale, measured on a five-point Likert scale (1—completely disagree, 5—completely agree). An example item is ‘I am sure that I would remember to use a condom although I have consumed alcohol or other drugs’. A higher score indicates higher perceptions of condom self-efficacy. Wider research shows it has a lower but acceptable internal reliability (α = 0.71; [36]).

### 2.14. PrEP Self-Efficacy

The PrEP self-efficacy behaviour subscale was used ([90]). This contains eight items such as ‘how difficult would it be for you to visit a doctor who can provide PrEP?’ measured on a four-point Likert scale (1—very hard to do, 4—very easy to do). A higher score indicates a higher PrEP self-efficacy. Previous related research shows this to have good internal reliability (α = 0.87; [34]).

### 2.15. PrEP Acceptability

The Attitudes towards PrEP Scale ([53]) is 14-item measure of PrEP acceptability, measured on a five-point Likert scale (1—strongly disagree, 5—strongly agree). Example items include ‘the NHS should fund PrEP’ and ‘PrEP is likely to work’. A higher score indicates higher PrEP acceptability. The validation of this scale showed a similarly acceptable internal reliability (α = 0.79; [53]).

### 2.16. Data Analyses

Sample characteristics and descriptive statistics were conducted in SPSS 26. These included the mean, standard deviation, and internal validity for each variable. Bivariate correlations were also analysed. The Latent Profile Analysis (LPA) was conducted using R Studio (Version: 2025.05.1+513) and the package ‘TidyLPA’ ([78]). Statistical recommendations for model selection were based on several fit indices ([46]), namely lower Bayesian information criterion (BIC) and Akaike information criterion (AIC) scores, an Entropy score closer to 1 (range 0–1), and a significant (i.e., *p* ≤ 0.05) bootstrapped likelihood ratio test (BLRT_p). Once the model had been selected, follow up analyses were conducted as recommended by the ‘gold standard’ three-step method for LPA, analysis of variance ([84]). Analysis of variance (ANOVA) with pairwise comparisons (adjusted with the Bonferroni correction) was conducted for each variable across the chosen number of latent profiles.

## 3. Results

The age range for the total sample was between 18 and 73 years (*M* = 35.61, *SD* = 9.95), with a split of *n* = 209 PrEP users and *n* = 291 Non-PrEP Users. A majority of the sample were White (91.2%), gay (73.2%) and educated to undergraduate degree level (39.2%). A breakdown of all the demographic data can be found in (Table 1).

Descriptive statics are provided in (Table 2), and all variables used had an acceptable internal reliability.

Pearson’s correlations are provided in (Table 3). It is noteworthy that PrEP acceptability was moderately positively correlated with HIV stigma (r = 0.55, *p* ≤ 0.01) and LGBTQ+ connectedness (r = 0.42, *p* ≤ 0.01), and negatively correlated with trust in science (r = −0.47, *p* ≥ 0.01). However, while perceived risk of HIV was weakly negatively correlated with condom self-efficacy (r = −0.19, *p* ≥ 0.01), it was not significantly correlated with risky sexual behaviour or PrEP acceptability.

### Latent Profile Analysis

The Latent Profile Analysis (LPA) yielded four distinct profiles of MSM in relation to PrEP acceptability. Fit parameters for latent profiles 1–5 estimated from the entire data set are presented in (Table 4). While the model reasonably supported both a three- and four-profile solution, thorough inspection of the groups allows for smaller but meaningful groups to be considered ([79]). The four-profile solution was justifiable in line with current PrEP literature ([14]; [19]; [86]) and subsequently chosen.

Figure 1 shows the fit parameters for profiles 1–4 across the breadth of the main analytic variables.

One-way between-groups ANOVA were conducted to analyse group differences across the four latent profiles. Post hoc analyses for any significant results were conducted using pairwise comparisons with Bonferroni correction, set to a *p* value of 0.0125. Results of the ANOVA and the follow up tests are presented in (Table 5). The classes were then qualitatively defined.

Class 1 (*n* = 75, 15%) was labelled ‘PrEP Ambivalent’—participants indicated high levels of PrEP acceptability, as well as high levels of condom self-efficacy, identity resilience and low perceived risk of HIV. Interestingly, while levels of HIV stigma were low, the likelihood of having sex with a HIV+ undetectable partner was also very low. Significant differences in sociosexual orientation may also indicate lower levels of casual sex with multiple partners. As such, it is possible that participants within this profile have favourable attitudes but do not perceive a need for it at this time ([88]).

Class 2 (*n* = 181, 36.2%) was labelled ‘PrEP Accepting’—participants had on average the highest levels of PrEP acceptability, risky sexual behaviour, and LGBTQ+ connectedness. They had lower HIV stigma than classes 3 and 4, and most likely to have sex with someone who is HIV+ undetectable. This class had the highest percentage of PrEP users among them (70.7%) and it is likely that these participants would have high perceived PrEP candidacy ([94]).

Class 3 (*n* = 185, 37%) was labelled ‘PrEP Hesitant’—participants indicated lower levels of PrEP acceptability but lower condom self-efficacy and higher levels of risky sexual behaviour than classes 2 and 4. Interestingly, participants had lower levels of internalised homonegativity and high levels of LGBTQ+ connectedness. They had more positive attitudes towards the NHS but lower trust in science and PrEP self-efficacy. These participants may be hesitant to engage with PrEP because of levels of medical mistrust, side-effect concerns, or wider systemic barriers ([8]; [16]).

Class 4 (*n* = 59, 11.8%) was labelled ‘PrEP Rejecting’—participants had the lowest levels of PrEP acceptability, trust in science and LGBTQ+ connectedness and highest levels of internalised homonegativity, and HIV stigma. While *N* = 9 of participants in this group were PrEP users, it is likely that this class of individuals represents those who would reject PrEP based on identity factors or fear of association of PrEP and the LGBTQ+ community ([4]; [50]).

## 4. Discussion

This study sought to investigate how various psychosocial factors (e.g., internalised homonegativity; trust in science, condom self-efficacy) influenced PrEP acceptability among men who have sex with men (MSM) in the United Kingdom (UK). From our latent profile analysis, four classes were quantitatively identified and qualitatively defined in line with wider literature. These were defined as (1) PrEP Ambivalent; (2) PrEP Accepting; (3) PrEP Hesitant; and (4) PrEP Rejecting.

Consistent with existing research, we found that various psychosocial constructs could impact PrEP acceptability. Perceptions of HIV risk and stigma ([43]), sexual behaviours ([33]), aspects of social identity ([4]), and social values ([54]) all contributed to differing levels of PrEP acceptability. However, importantly, we have identified heterogeneity across profiles of MSM, providing insight into barriers to PrEP uptake in a group of individuals who may be at increased need of HIV prevention ([27]). By demonstrating the differentiating subgroups of MSM in relation to PrEP acceptability based on patterns of psychosocial differences, our results suggest useful ways to improve initiatives for PrEP uptake across the UK.

The **PrEP Ambivalent** group represented 15% of the sample and while having high levels of PrEP acceptability had low levels of PrEP usage with 76% (*n* = 57) not using PrEP. The high levels of condom self-efficacy observed in this cohort suggest that condoms may be the preferred method of HIV prevention and that they may therefore have a lower need for PrEP ([69]). HIV stigma was low within this profile; however, as was their likelihood of engaging in condomless sex with a partner who is HIV+ and undetectable. These findings may point to higher levels of “serosorting,” where individuals mitigate their risk of HIV by selectively choosing partners they perceive to be at minimal or no risk ([17]; [25]). Additionally, the relatively low sociosexual orientation scores suggest a pattern of fewer sexual partners, which, while reflecting an overall acceptance of PrEP, may indicate that these individuals do not perceive a significant need for a biomedical prevention strategy ([22]). This is further supported by the low levels of perceived HIV risk, which would logically reduce the perceived necessity for PrEP use ([49]).

The **PrEP Accepting profile**, representing the second-largest group, also exhibited the highest percentage of PrEP users (*N =* 128). This group may reflect individuals who perceive a higher risk of HIV, thereby driving a self-perceived need for PrEP ([82]). This inclination towards PrEP use could be influenced by factors such as lower condom self-efficacy, higher engagement in risky sexual behaviours, and/or greater sociosexual orientation. Notably, while this group demonstrated high levels of trust in scientific evidence, they expressed a lack of confidence in the ability of the NHS to adequately serve their community. This discrepancy may be attributed partly to the political dynamics surrounding PrEP access and implementation ([66]). Additionally, their higher likelihood of engaging in unprotected sex with an HIV+ partner with an undetectable viral load suggests a stronger belief in Treatment as Prevention (TasP) ([2]), a finding that contradicts previous research ([21]).

The **PrEP Hesitant** group, which emerged as the largest in this sample (*n* = 185, 37%), is characterised by low levels of PrEP acceptability. This group may reject PrEP due to social stigmas associated with its use, such as perceptions of promiscuity or irresponsibility ([16]; [23]; [83]). Furthermore, this group expressed relatively positive attitudes towards the NHS, suggesting that their reluctance to adopt PrEP may stem from concerns about its potential negative impact on social healthcare systems ([45]). Additionally, higher levels of internalised homonegativity could predispose this group to welfare chauvinism, where they might be less supportive of public health interventions perceived to be linked with marginalised groups ([13]). While the proportion of PrEP users within this group was small (*n* = 54, 29.2%), it is crucial to acknowledge that PrEP use is not always openly disclosed, meaning that some individuals may still be using PrEP in secret, potentially reinforcing feelings of shame and further complicating their acceptance of the drug ([38]; [47]).

The smallest group in this sample, **PrEP Rejecting**, displayed notable differences from the other profiles. This group had the lowest levels of both PrEP acceptability and trust in science, as well as diminished identity resilience and weaker connections to the LGBTQ+ community. They also exhibited the highest levels of internalised homonegativity and HIV stigma. These characteristics suggest that this group may reject PrEP due to the stigma they associate with both HIV and their sexual orientation ([8]; [12]). Additionally, the lower levels of identity resilience observed in this group may make them less likely to adopt protective health behaviours, which could contribute to poorer health outcomes in the future ([47]; [50]).

Taken together, these findings suggest the need for differentiated intervention strategies tailored to the psychosocial profiles identified. For example, regarding the PrEP Ambivalent group, efforts may focus on reinforcing condom-based prevention while gently introducing PrEP as a flexible option for future risk contexts (e.g., events-based PrEP) ([88]). The PrEP Accepting group would benefit from ongoing support to sustain engagement and prevent PrEP fatigue, such as regular check-ins with trusted healthcare providers, peer-led support models, and broader structural improvements to NHS trust and access ([59]). For the PrEP Hesitant group, interventions should seek to challenge PrEP-related stigma and misconceptions, possibly through community-tailored narratives, trusted community champions, in order to increase sexual identity positivity ([47]). Additionally, future research could examine the relationship between self-worth, and acceptance of sexual identity, to better understand how these dimensions of identity influence PrEP decision-making, particularly in the context of intersecting identities (e.g., [26]). The PrEP Rejecting group may benefit from more radical and upstream interventions, including access to PrEP through general practice or pharmacies rather than traditional sexual health clinics, which can be stigmatising or inaccessible—particularly for MSM who do not identify as gay or bisexual ([64]). Creating non-judgemental, identity-affirming spaces outside of LGBTQ+ specific services could help rebuild trust and support biomedical engagement ([41]).

Overall, the acceptability of PrEP is just one component within a complex web of factors influencing the decision to use it ([39]; [53]), and it is crucial to address the disconnect between behavioural intentions and actual behaviours ([35]; [51]). This study, however, employs ‘person-centred’ empirical methods ([89]) to explore the intricate interplay of psycho-social influences that may prevent individuals with a recognized “PrEP need” from accessing this vital biomedical intervention ([69]). The results of this study should inform policy decisions regarding PrEP implementation in the UK. Only through a deeper, post hoc understanding of the factors influencing the decision to use PrEP can tailored and effective interventions begin to address the widening ‘PrEP Gap’ ([19]). Future research should adopt a more targeted approach, as discussed, in developing interventions to reduce both PrEP- and HIV-related stigma. Additionally, exploring demographic differences would provide valuable insights, though this was beyond the scope of the present study. Finally, it is well established that adherence to public health interventions can deteriorate over time (e.g., [63]), as seen in analogous pandemics such as COVID-19, where pandemic fatigue contributed to reduced engagement with preventive measures (e.g., [73]). Similar trends have been observed in declining measles vaccine uptake due to diminished trust in science (e.g., [85]). However, HIV remains a complex and long-standing pandemic with a unique social stigma, and our findings shed new light on additional psychosocial barriers to PrEP uptake. Nonetheless, longitudinal research is needed to validate these findings and to explore strategies to sustain PrEP adherence over time.

This study has several limitations. While it triangulates existing research and theory, Latent Profile Analysis (LPA) is inherently prone to identifying ‘spurious’ profiles ([84]) and is more suited for hypothesis generation rather than hypothesis testing. Additionally, the use of cross-sectional data, while offering some predictive value, does not provide insight into causation. Although the psychosocial variables were selected based on strong theoretical and empirical grounding, they were researcher-determined. As such, there is an inherent subjectivity in variable selection that may have influenced the types of profiles identified. This is a common consideration in person-centred analyses, where the interpretation of results is shaped, in part, by the constructs included ([68]). However, the profiles identified in this study align with patterns reported in previous research using these methodologies (e.g., ([84])). This provides some reassurance that the profiles observed here are consistent with broader trends in the literature. The sample was also majority white and highly educated and, thus, future research should aim to use stratified sampling methods to assess PrEP acceptability differences in other marginalised groups that are also at increased risk of HIV acquisition. It is crucial to acknowledge that psychological barriers to PrEP uptake do not operate in isolation, and future research from a broader public health perspective should incorporate structural factors (e.g., accessibility of specialised sexual health clinics) and community-level factors (e.g., PrEP education and health communications) to avoid reproducing a deficit model ([39]; [80]).

Finally, given the shifting socio-political terrain ([40]), it is vital that future interventions not only address structural and psychological barriers to PrEP uptake but also critically engage with the evolving forms of stigma and resilience among MSM. Understanding how these factors manifest in today’s climate can help ensure that public health responses are not only evidence-based but also culturally and politically responsive ([55]). Research increasingly shows that distrust in science and medical systems is often intertwined with broader political beliefs and attitudes, such as conservative or populist ideologies, which can shape perceptions of health interventions (e.g., COVID-19 Prevention; [42]). This intersection is particularly relevant to PrEP uptake, as lower trust in science and more conservative political attitudes have been associated with greater hesitancy to adopt biomedical prevention methods (e.g., [15]). Recognising these intersections is crucial for developing public health messaging that is not only evidence-based but also culturally and politically responsive, particularly for subgroups who may be hesitant or rejecting of PrEP. This study offers an empirically grounded starting point for such work.

## 5. Conclusions

In conclusion, this study identifies four distinct profiles of MSM in relation to PrEP acceptability, alongside other psychosocial factors influencing sexual health behaviours. By understanding the key differences among these groups, the likelihood of PrEP use can be better predicted and addressed through targeted public health interventions. For those groups with low levels of PrEP acceptability—making them less likely to use or disclose PrEP usage—stigma remains a significant barrier that must be tackled. Additionally, trust in science and attitudes towards the NHS emerge as crucial factors, contributing to PrEP decision-making and should be incorporated into future intervention strategies.

## Figures and Tables

**Figure 1 behavsci-15-00818-f001:**
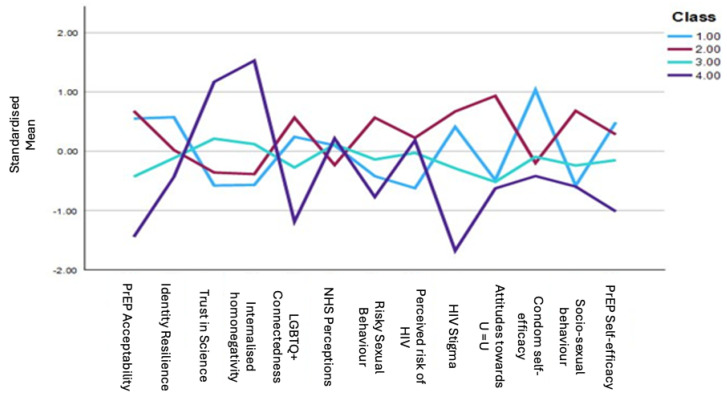
PrEP Acceptability profiles using standardised means.

**Table 1 behavsci-15-00818-t001:** Sociodemographic characteristics of total sample set and between latent profile groups.

Demographics	Total	Latent Profiles			
		1	2	3	4
	(N = 500)	(N = 75)	(N = 181)	N = (185)	(N = 59)
**Current PrEP Use**					
Using or used PrEP	41.8% (n = 209)	24.0% (n = 18)	70.7% (n = 128)	29.2% (n = 54)	15.3% (n = 9)
Never used PrEP	58.2% (n = 291)	76.0% (n = 57)	29.3% (n = 53)	70.8% (n = 131)	84.7% (n = 50)
**PrEP Dosage**					
Daily PrEP	19.8% (n = 99)	6.7% (n = 5)	38.7% (n = 70)	10.8% (n = 20)	6.8% (n = 4)
Event-Based PrEP	13.2% (n = 66)	6.7% (n = 5)	21.5% (n = 39)	10.3% (n = 19)	5.1% (n = 3)
No longer using PrEP	7.6% (n = 38)	10.7% (n = 8)	7.7% (n = 14)	7.6% (n = 14)	3.4% (n = 2)
Other	1.2% (n = 6)	0.0% (n = 0)	2.8% (n = 5)	0.5% (n = 1)	0.0% (n = 0)
Never used PrEP	58.2% (n = 291)	76.0% (n = 57)	29.3% (n = 53)	70.8% (n = 131)	84.7% (n = 50)
**Relationship Status**					
Single	45.8% (n = 229)	32% (n = 24)	56.9% (n = 103)	39.5% (n = 73)	49.2% (n = 29)
In a relationship (monogamous)	31.4% (n = 157)	49.3% (n = 37)	11.6% (n = 21)	42.7% (n = 79)	33.9% (n = 20)
In a relationship (non-monogamous)	22.4% (n = 112)	18.7% (n = 14)	31.5% (n = 57)	17.3% (n = 32)	15.3% (n = 9)
Other	0.4% (n = 2)	0.0% (n = 0)	0.0% (n = 0)	0.5% (n = 1)	1.7% (n = 1)
**Sexuality**					
Gay	73.2% (n = 366)	74.7% (n = 56)	83.4% (n = 151)	69.7% (n = 129)	50.8% (n = 30)
Bisexual	25% (n = 125)	24.0% (n = 18)	13.3% (n = 24)	29.2% (n = 54)	49.2% (n = 29)
Other	1.8% (n = 9)	1.3% (n = 1)	3.3% (n = 6)	1.1% (n = 2)	0.0% (n = 0)
**Gender Identity**					
Same as birth (i.e., Cisgender Male)	98.4% (492)	98.7% (n = 74)	96.7% (n = 175)	99.5% (n = 184)	100% (n = 59)
Different from birth (i.e., Non-Binary)	1.6% (n = 8)	1.3% (n = 1)	3.3% (n = 6)	0.5% (n = 1)	0.0% (n = 0)
**Outness**					
Out to everyone!	73% (n = 365)	77.3% (n = 58)	86.7% (n = 157)	67.6% (n = 125)	42.4% (n = 25)
Not out at all!	3% (n = 15)	0.0% (n = 0)	0.6% (n = 1)	2.7% (n = 5)	15.3% (n = 9)
Out to some people!	24% (n = 120)	22.7% (n = 17)	12.7% (n = 23)	29.7% (n = 55)	42.4% (n = 25)
**Ethnicity**					
White	91.2% (n = 456)	90.7% (n = 68)	91.7% (n = 166)	94.1% (n = 174)	81.4% (n = 48)
Mixed	2.6% (n = 13)	4.0% (n = 3)	2.2% (n = 4)	1.6% (n = 3)	5.1% (n = 3)
Asian/British Asian	3.4% (n = 17)	1.3% (n = 1)	2.8% (n = 5)	3.2% (n = 6)	8.5% (n = 5)
Black	1.4% (n = 7)	2.7% (n = 2)	1.7% (n = 3)	0.0% (n = 0)	3.4% (n = 2)
Other	1.4% (n = 7)	1.3% (n = 1)	1.7% (n = 3)	1.1% (n = 2)	1.7% (n = 1)
**Marital Status**					
Never married	75.2% (376)	74.7% (n = 56)	79.6% (n = 144)	71.9% (n = 133)	72.9% (n = 43)
Married	19.2% (n = 96)	20.0% (n = 15)	15.5% (n = 28)	21.6% (n = 40)	22.0% (n = 13)
Divorced	3.8% (n = 19)	2.7% (n = 2)	4.4% (n = 8)	4.3% (n = 8)	1.7% (n = 1)
Widowed	1.0% (n = 5)	0.0% (n = 0)	0.6% (n = 1)	1.6% (n = 3)	1.7% (n = 1)
Other	0.8% (n = 4)	2.7% (n = 2)	0.0% (n = 0)	0.5% (n = 1)	1.7% (n = 1)
**Education**					
GCSE	5.4% (n = 27)	2.7% (n = 2)	1.7% (n = 3)	8.1% (n = 15)	11.9% (n = 7)
A level	17.4% (n = 87)	14.7% (n = 11)	12.7% (n = 23)	20.5% (n = 38)	25.4% (n = 15)
Undergraduate	39.2% (n = 196)	42.7% (n = 32)	35.4% (n = 64)	42.2% (n = 78)	37.3% (n = 22)
Postgraduate	26.8% (n = 134)	33.3% (n = 25)	33.1% (n = 60)	19.5% (n = 36)	22.0% (n = 13)
PhD/Prof Doc	11.2% (n = 56)	6.7% (n = 5)	17.1% (n = 31)	9.7% (n = 18)	3.4% (n = 2)
**UK Location**					
England	84.2% (n = 421)	80% (n = 60)	85.6% (n = 155)	83.2% (n = 154)	88.1% (n = 52)
Scotland	9.8% (n = 49)	8.0% (n = 6)	9.9% (n = 18)	10.3% (n = 19)	10.2% (n = 6)
Wales	4.0% (n = 20)	8.0% (n = 6)	2.2% (n = 4)	4.9% (n = 9)	1.7% (n = 1)
Northern Ireland	2.0% (n = 10)	4.0% (n = 3)	2.2% (n = 4)	1.6% (n = 3)	0.0% (n = 0)
**Age**	Aged 18–73	Aged 20–73	Aged 18–58	Aged 21–66	Aged 18–64
	(*M* = 35.61, *SD* = 9.95)	(*M* = 34.4, *SD* = 11.02)	(*M* = 34.02, *SD* = 7.97)	(*M* = 36.85, *SD* = 10.33)	(M = 37.53, *SD* = 12.0)

**Table 2 behavsci-15-00818-t002:** Descriptive statistics for each of the psychosocial variables.

Total Sample (N = 500)	Mean	SD	α	95% Confidence Boundaries	
				Lower	Upper
Perceived risk of HIV	2.81	0.67	0.8	0.78	0.83
HIV stigma	4.33	0.52	0.91	0.9	0.93
Condom use self-efficacy	3.73	0.65	0.7	0.66	0.74
Identity resilience	3.51	0.52	0.84	0.81	0.86
LGBTQ+ connectedness	2.76	0.82	0.98	0.97	0.98
NHS Perceptions	2.27	0.64	0.84	0.82	0.87
Trust in science	1.92	0.55	0.89	0.88	0.91
Risky sexual behaviour	2.94	0.77	0.68	0.64	0.72
Socio-sexual behaviour	5.71	1.57	0.86	0.85	0.88
Internalised homonegativity	1.59	0.73	0.83	0.8	0.85
Attitudes towards U = U	3.13	1.89	Single item measure		
PrEP self-efficacy	3.02	0.54	0.78	0.75	0.81
PrEP acceptability	3.79	0.46	0.8	0.77	0.82

**Table 3 behavsci-15-00818-t003:** Correlation matrix among the main analytic variables (N = 500).

		1	2	3	4	5	6	7	8	9	10	11	12
1	Perceived risk of HIV	1											
2	HIV stigma	0.04	1										
3	Condom self-efficacy	−0.19 **	0.10 *	1									
4	Identity resilience	−0.23 **	0.09	0.25 **	1								
5	LGBTQ+ Connectedness	0.03	0.40 **	0.04	0.18 **	1							
6	NHS attitudes	−0.14 **	−0.12 **	0.17 **	10 *	0.65	1						
7	(Mis)Trust in science	0.09	−0.41 **	−0.24 **	−0.18 **	−0.29 **	0.02	1					
8	Risky sexual behaviour	0.06	0.23 **	−0.19 **	<0.01	0.23 **	−0.10 *	−0.11 *	1				
9	Sociosexual orientation	0.26 **	0.31 **	−0.14 **	0.02	0.19 **	−0.05	−0.12 **	0.49 **	1			
10	Internalised homonegativity	0.14 **	−0.42 **	−0.22 **	−0.28 **	−0.31 **	−0.03	0.23 **	−0.18 **	−0.17 **	1		
11	Sex with HIV+ (undetectable)	0.08	0.38 **	−0.17 **	< 0.01	0.29 **	−0.15 **	−0.14 **	0.37 **	0.34 **	−0.21 **	1	
12	PrEP self-efficacy	−0.08	0.25 **	0.33 **	0.25 **	0.20 **	0.17 **	−0.25 **	0.17 **	0.20 **	−0.31 **	0.18 **	1
13	PrEP acceptability	−0.04	0.55 **	0.12 *	0.14 **	0.42 **	−0.17 **	−0.47 **	0.22 **	0.23 **	−0.32 **	0.38 **	0.35 **

** Correlation is significant at the (*p* = 0.01) level (2-tailed) * Correlation is significant at the (*p* = 0.05) level (2-tailed).

**Table 4 behavsci-15-00818-t004:** Fit parameters for latent profiles 1–5.

Model	Classes	AIC	BIC	Entropy	Prob_min	Prob_max	n_min	n_max	BLRT_p
1	1	18,485.19	18,594.77	1	1	1	1	1	
1	2	17,807.68	17,976.26	0.78	0.91	0.95	0.35	0.65	0.01
1	3	17,475.46	17,703.05	0.87	0.93	0.95	0.13	0.45	0.01
**1**	**4**	**17,391.57**	**17,678.17**	**0.8**	**0.75**	**0.94**	**0.12**	**0.37**	**0.01**
1	5	17,360.68	17,706.28	0.77	0.73	0.96	0.1	0.36	0.01

**Table 5 behavsci-15-00818-t005:** Means, standard deviations, ANOVA and post hoc comparisons for each of the analytic variables across the 4 profiles to show heterogenous differences.

Profile	(1) PrEP Ambivalent		(2) PrEP Accepting		(3) PrEP Hesitant		(4) PrEP Rejecting		ANOVA		Post hoc Bonferroni Correction ^†^			
	M	SD	M	SD	M	SD	M	SD	*F* (3, 496)	η2	1	2	3	4
											Significant difference:			
Perceived risk of HIV	2.4	(0.63)	2.96	(0.62)	2.79	(0.62)	2.94	(0.9)	14.64 **	0.08	2,3,4	1	1	1
HIV stigma	4.54	(0.38)	4.67	(0.19)	4.18	(0.39)	3.46	(0.51)	203.89 **	0.55	4	3,4	1,2,4	1,2,3
Condom self-efficacy	4.4	(0.47)	3.6	(0.62)	3.67	(0.57)	3.46	(0.63)	41.08 **	0.2	2,3,4	1	1	1
Identity resilience	3.8	(0.48)	3.52	(0.49)	3.45	(0.5)	3.28	(0.56)	13.71 **	0.08	2,3,4	1,4	1	1,2
LGBTQ+ Connectedness	2.94	(0.75)	3.22	(0.59)	2.4	(1.04)	1.35	(1.3)	72.06 **	0.3	3,4	3,4	1,2,4	1,2,3
NHS attitudes	2.34	(0.6)	2.13	(0.64)	2.35	(0.62)	2.41	(0.65)	5.4 **	0.03		3	2	
(Mis)Trust in science	1.6	(0.42)	1.72	(0.48)	2.04	(0.43)	2.57	(0.63)	62.92 **	0.28	3,4	3,4	1,2,4	1,2,3
Risky sexual behaviour	2.62	(0.68)	3.38	(0.64)	2.83	(0.7)	2.35	(0.74)	46.7 **	0.22	2	1,3,4	2,4	2,3
Sociosexual orientation	4.78	(0.23)	6.78	(1.27)	5.33	(1.38)	4.77	(1.5)	65.12 **	0.28	2	1,3,4	2	2
Internalised homonegativity	1.17	(0.35)	1.3	(0.44)	1.67	(0.62)	2.71	(0.93)	103.05 **	0.38	3,4	3,4	1,2,4	1,2,3
Sex with HIV+ (undetectable)	2.21	(0.48)	4.9	(1.16	2.16	(0.62)	1.95	(1.44)	163.77 **	0.5	2	1,3,4	2	2
PrEP self-efficacy	3.28	(0.5)	3.17	(0.48)	2.94	(0.47)	2.48	(0.5)	40.11 **	0.2	3,4	3,4	1,2,4	1,2,3
PrEP acceptability	4.04	(0.34)	4.1	(0.31)	3.59	(0.31)	3.12	(0.35)	184.86 **	0.53	3,4	3,4	1,2,4	1,2,3

^†^ Mean difference significant at 0.0125 level; ** *p* ≤ 0.001.

## Data Availability

Limited data can be supplied upon reasonable request to the corresponding author.

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
