# Peer review of "Psychosocial Profiles of Men Who Have Sex with Men (MSM) Influencing PrEP Acceptability: A Latent Profile Analysis"

_behavsci, 2025, doi:10.3390/bs15060818_

Round 1

Reviewer 1 Report

Comments and Suggestions for Authors

This is an excellent article. It is meticulously researched and persuasively argued. I see no need for further changes and it is, in my opinion, ready for publication. Invaluable and timely work indeed.

Author Response

1a. This is an excellent article. It is meticulously researched and persuasively argued. I see no need for further changes and it is, in my opinion, ready for publication. Invaluable and timely work indeed

Response: Thank you so much for taking the time to review this article and we are delighted with the positive response. We have made amendments to the manuscript overall based on other reviewer feedback, and hope this contributes positively to the work overall.

Reviewer 2 Report

Comments and Suggestions for Authors

1.- Originality

The article addresses a relevant and current topic: the relationship between sexuality and health. It is timely, as it again sheds light on how moral issues trump health and how PrEP research needs to detach itself from MSM groups. There seems to be a revived war on sex in recent years, and anyone who is not heterosexual seems to be once again the target of control. Page 2 is strong in this sense, and lines 79-98 are thought-provoking.

2.-Methodology

The methodology is perfect for this study. As a qualitative researcher, I appreciate the information delivered by a method that gathers information from 500 participants. The only measure I would like in future research is self-worth, which tells us something different from self-esteem line 128. Always remember that criminals have high self-esteem, too.

3.- Strength

The strength is how the results are presented. These inform the readers on how mistrust in science, and sometimes with good reason, makes PrEP such a hard sell (292-287). The side effects need to be made clear in lay terms. Moreover, the population is once again publicly linked to HIV.

Lines 400-404 do not describe a limitation; they describe a strength. It is impossible to relate to an unmediated reality because the results and our analyses are always intersubjective. Our values and beliefs are reflected in our analysis and are evident in how we interpret the data (Tremblay 2021).

The discussion is filled with great insights. As someone who researches stigmatised and marginalised populations, it held my attention from beginning to end. The discussion is often the weakest part in quantitative studies – not here.

4.- Weaknesses

I do not see any.

5.- Recommendations for the discussion and conclusion:

Future research should explore and emphasise the connections between public health, the role of medical institutions and stigma. Plus, a focus on self-worth is the point of departure of health.

Author Response

2a. The article addresses a relevant and current topic: the relationship between sexuality and health. It is timely, as it again sheds light on how moral issues trump health and how PrEP research needs to detach itself from MSM groups. There seems to be a revived war on sex in recent years, and anyone who is not heterosexual seems to be once again the target of control. Page 2 is strong in this sense, and lines 79-98 are thought-provoking.

Response: Thank you for this feedback, and for taking the time to review this manuscript. We agree work like this is hugely important during this current socio-political climate and are happy it is contributing positively to this cause.

2b. The methodology is perfect for this study. As a qualitative researcher, I appreciate the information delivered by a method that gathers information from 500 participants. The only measure I would like in future research is self-worth, which tells us something different from self-esteem line 128. Always remember that criminals have high self-esteem, too.

Response: 

Thank you for this feedback. While we included identity resilience (incl. self-esteem), we agree that self-worth could capture distinct relational and psychosocial dimensions relevant to PrEP decision-making and have articulated this in our discussion section:

“Additionally, future research could examine the relationship between self-esteem, self-worth, and acceptance of sexual identity, to better understand how these dimen-sions of identity influence PrEP decision-making, particularly in the context of inte-secting identities (e.g., Elopre et al., 2021)” (Lines 399-401)

2c. The strength is how the results are presented. These inform the readers on how mistrust in science, and sometimes with good reason, makes PrEP such a hard sell (292-287). The side effects need to be made clear in lay terms. Moreover, the population is once again publicly linked to HIV.

Response: Thank you for this positive feedback.

2d. Lines 400-404 do not describe a limitation; they describe a strength. It is impossible to relate to an unmediated reality because the results and our analyses are always intersubjective. Our values and beliefs are reflected in our analysis and are evident in how we interpret the data (Tremblay 2021).

Response: We appreciate the reviewer’s perspective, and we agree that this can also be viewed as a strength of the approach. However, our intention in lines 400–404 was to highlight the importance of acknowledging this subjectivity explicitly, particularly for readers who may be more accustomed to a realist or positivist paradigm. Therefore, we believe that presenting this as a “limitation” remains appropriate within the context of our chosen research design and for the broader academic readership.

2e. The discussion is filled with great insights. As someone who researches stigmatised and marginalised populations, it held my attention from beginning to end. The discussion is often the weakest part in quantitative studies – not here.

Response: We thank you again for your positive feedback.  

2f. Future research should explore and emphasise the connections between public health, the role of medical institutions and stigma. Plus, a focus on self-worth is the point of departure of health.

Response:  Thank you for suggesting this. We have aimed to do this in line with other reviewer feedback also:

(e.g., Additionally, future research could examine the relationship between self-worth, and acceptance of sexual identity, to better understand how these dimensions of identity influence PrEP decision-making, particularly in the context of intersecting identities (e.g., Elopre et al., 2021); Lines 415-418).

Reviewer 3 Report

Comments and Suggestions for Authors

This manuscript provides intriguing, important, and exciting research into understanding the psychological predictors of PrEP acceptability in men who have sex with men. The public health impact of these findings are significant and important, and provide deeper knowledge and data to better understand those psychological factors that enhance or limit usage of PrEP medications as a form of prevention of HIV transmission. This manuscript is of very high quality and is very well written, with important and logical conclusions drawn from very sound data. The following brief comments are provided to further enhance the quality of this manuscript:

Introduction

  • Please consider providing a brief overview of the different kinds of PrEP medications and modes of delivery – i.e. daily oral medications such as Truvada and Descovy, and injectable medications like Apretude that are delivered a few times a year.
  • Many are still not aware of the incredible effectiveness of PrEP mediations in preventing transmission to HIV-negative individuals. Please consider providing a few sentences that describe the effectiveness of these medications of preventing transmission.

Methods

  • The methods section is excellent, and the breadth of instruments used to collect data is impressive.
  • But why these specific instruments? What was it about the data that would be provided by each that connected so importantly to the research questions being posed. Please consider adding a few sentences (perhaps a paragraph at most) to section 2.2 to explain why the following 13 instruments were selected for this study, just to provide a bit of description as to why these scales were so well aligned individually and collectively towards addressing the research questions.

Discussion

  • The results of this study are fascinating and are of tremendous importance to the field of public health.
  • One area that is of tremendous importance that I personally gleaned from this manuscript is how reduced trust in science impacts PrEP usage. And often this distrust in science is rooted in one’s political beliefs/attitudes, which can have far reaching effects. Study authors may wish to provide a bit more detail on the reflection section about research/literature in the field that indicates how one’s political beliefs and attitudes impact health decisions and behaviors, as this would not only give further insight into what can influence psychological beliefs and attitudes towards PrEP, but also how to craft better health communication messages towards those that are hesitant or rejecting of PrEP.  

Author Response

3a. This manuscript provides intriguing, important, and exciting research into understanding the psychological predictors of PrEP acceptability in men who have sex with men. The public health impact of these findings are significant and important, and provide deeper knowledge and data to better understand those psychological factors that enhance or limit usage of PrEP medications as a form of prevention of HIV transmission. This manuscript is of very high quality and is very well written, with important and logical conclusions drawn from very sound data. The following brief comments are provided to further enhance the quality of this manuscript:

Response: We sincerely thank the reviewer for their thoughtful and generous comments on our manuscript. We also welcome the reviewer’s helpful suggestions and are committed to integrating them to further enhance the clarity and impact of the manuscript.

3b. Please consider providing a brief overview of the different kinds of PrEP medications and modes of delivery – i.e. daily oral medications such as Truvada and Descovy, and injectable medications like Apretude that are delivered a few times a year.

Response: We thank the reviewer for their helpful suggestion. We have incorporated a brief description of how PrEP is typically administered in the UK, noting that it is available free at the point of access and is most commonly taken as a daily oral medication comprising emtricitabine and tenofovir disoproxil. Given the context of the UK sample and the focus of this manuscript, we have refrained from discussing alternative PrEP delivery methods such as long-acting injectables at this stage, as they have yet to be commissioned within the NHS and may risk introducing confusion regarding the accessibility and current status of PrEP provision in the UK. We believe that maintaining focus on the current standard of care ensures clarity and relevance for the reader.

“PrEP is currently available in the UK free at the point of access, administered as an oral medication of emtricitabine/tenofovir disoproxil and typically taken daily (Brady et al., 2019; Kirby, 2020)” (Lines 41-43)

3c. Many are still not aware of the incredible effectiveness of PrEP mediations in preventing transmission to HIV-negative individuals. Please consider providing a few sentences that describe the effectiveness of these medications of preventing transmission.

Response: We agree entirely and have now elaborated this, including evidence from UK clinical trials: “Clinical trials, including the PROUD study conducted in the UK, have demonstrated that oral PrEP is highly effective at reducing HIV acquisition among populations at in-creased risk of HIV acquisition (Donnell et al., 2014; McCormack et al., 2016b)” (Lines 43-45).

3d. The methods section is excellent, and the breadth of instruments used to collect data is impressive. But why these specific instruments? What was it about the data that would be provided by each that connected so importantly to the research questions being posed. Please consider adding a few sentences (perhaps a paragraph at most) to section 2.2 to explain why the following 13 instruments were selected for this study, just to provide a bit of description as to why these scales were so well aligned individually and collectively towards addressing the research questions.

Response: We have now elaborated on this in a short paragraph as recommended in section 2.2 : “Psychometric measures were selected following the conventions of similar LPA studies in this area (e.g., Rzeszutek & GruszczyÅ„ska, 2020), after synthesising the literature on current psychosocial barriers to PrEP uptake. Additionally, context-specific research (e.g., Flowers et al., 2022; Gifford et al., 2025b) further highlighted key con-structs warranting additional exploration (i.e., internalised homonegativity, LGBTQ+ connectedness, trust in science) that align with our study’s aim of generating hypotheses about how these psychosocial factors influence PrEP acceptability. Each measure was therefore chosen for its theoretical relevance and empirical support in relation to our overarching research questions.”

3e. The results of this study are fascinating and are of tremendous importance to the field of public health. One area that is of tremendous importance that I personally gleaned from this manuscript is how reduced trust in science impacts PrEP usage. And often this distrust in science is rooted in one’s political beliefs/attitudes, which can have far reaching effects. Study authors may wish to provide a bit more detail on the reflection section about research/literature in the field that indicates how one’s political beliefs and attitudes impact health decisions and behaviors, as this would not only give further insight into what can influence psychological beliefs and attitudes towards PrEP, but also how to craft better health communication messages towards those that are hesitant or rejecting of PrEP.  

Response: We have reflected on this issue in relation to other analogous health research and further discussed its importance in relation to PrEP usage:

“Research increasingly shows that distrust in science and medical systems is often intertwined with broader political beliefs and attitudes, such as conservative or populist ideologies, which can shape perceptions of health interventions (e.g., Covid-19 Prevention; Hartmann & Müller, 2023). This intersection is particularly relevant to PrEP uptake, as lower trust in science and more conservative political attitudes have been associated with greater hesitancy to adopt biomedical prevention methods (e.g., Collins, 2022). Recognising these intersections is crucial for developing public health messaging that is not only evidence-based but also culturally and politically responsive, particularly for subgroups who may be hesitant or rejecting of PrEP.” (Lines 458-466).

Reviewer 4 Report

Comments and Suggestions for Authors

An interesting article discussing an important issue.  It is also a subject matter that has been now widely researched.  Therefore I would like to see the authors put forward a stronger reason for justifying the paper.

The conclusion could perhaps look at how other public health issues with time and as interventions and perceptions of risks are reduced also see lower adherence to interventions.  Is there anything different about the HIV story?  Is there new insights that your results can shed here?

What other factors should we be looking into to include in the model that predicts PrEP, tif we look at this issue with a large public health perspective?

Given your findings, are the current PrEP access programs getting to the right groups?  What recommendations can you suggest?

Author Response

4a. An interesting article discussing an important issue.  It is also a subject matter that has been now widely researched.  Therefore I would like to see the authors put forward a stronger reason for justifying the paper.

Response: We thank the reviewer for taking the time to review this paper, their comment(s), and for highlighting the importance of clearly justifying the rationale for this study. We have now emphasised the need for this research towards the end of the Introduction:

“While such factors influencing PrEP uptake have been widely researched, there remains a need for larger scale, quantitative, person-centred analyses in the UK, particularly in the context of free access to PrEP.” (Lines 107-109)

“This approach not only builds on existing research but also provides critical insights to inform tailored public health interventions and bridge the persistent PrEP Gap in the UK.” (Lines 114-117)

4b. The conclusion could perhaps look at how other public health issues with time and as interventions and perceptions of risks are reduced also see lower adherence to interventions.  Is there anything different about the HIV story?  Is there new insights that your results can shed here?

Response: Thank you for this comment. We have added a discussion point that considers analogous public health issues and how our findings compare and contrast with issues around maintaining adherence. We acknowledge that HIV is a highly complex pandemic with unique social stigmas, as discussed throughout the manuscript, and our findings shed light on these interplays of factors that could inform strategies for sustaining adherence. However, we note that longitudinal research is needed to build on these insights, which is beyond the scope of this current study but can certainly inform future research designs and directions:

“Finally, it is well established that adherence to public health interventions can deterio-rate over time (e.g., Middleton et al., 2013), as seen in analogous pandemics such as COVID-19, where pandemic fatigue contributed to reduced engagement with preven-tive measures (e.g., Petherick et al., 2021). Similar trends have been observed in declining measles vaccine uptake due to diminished trust in science (e.g., Torracinta et al., 2021). However, HIV remains a complex and long-standing pandemic with a unique social stigma, and our findings shed new light on additional psychosocial barriers to PrEP uptake. Nonetheless, longitudinal research is needed to validate these findings and to explore strategies to sustain PrEP adherence over time.” (Lines 440-448)

4c. What other factors should we be looking into to include in the model that predicts PrEP, tif we look at this issue with a large public health perspective?

Response: We received a similar comment from another reviewer and have discussed how self-worth may also be an interesting aspect of exploration for larger scale and longer-term research (see comment 2b). Furthermore, in the limitations section, acknowledged that from a broader public health perspective, future research should integrate psychological insights with structural and community-based factors that shape PrEP uptake:

“It is crucial to acknowledge that psychological barriers to PrEP uptake do not operate in isolation, and future research from a broader public health perspective should incorporate structural factors (e.g., accessibility of specialised sexual health clinics) and community-level factors (e.g., PrEP education and health communications) to avoid reproducing a deficit model (Golub et al., 2019b; Seethaler et al., 2019).” (Lines 463-467)

4d. Given your findings, are the current PrEP access programs getting to the right groups?  What recommendations can you suggest?

Response: We thank the reviewer for raising this important point. We agree that evaluating the current access, uptake, and success of PrEP programs in the UK is a critical public health issue. However, we respectfully note that this evaluation is beyond the scope of our study, which focused on identifying psychosocial profiles of PrEP acceptability rather than conducting a comprehensive programmatic assessment. Nevertheless, we acknowledge that systematic reviews (e.g., Coukan et al., 2023) have highlighted the need for improvements and tailored interventions within existing PrEP programs. Our study’s findings support this by recommending differentiated services (e.g., pharmacy-based PrEP access and targeted psychological support) that specifically address the needs of PrEP-hesitant and PrEP-rejecting groups, particularly regarding sexual identity positivity and trust in healthcare.